# Subclinical Acute Kidney Injury in COVID-19: Possible Mechanisms and Future Perspectives

**DOI:** 10.3390/ijms232214193

**Published:** 2022-11-17

**Authors:** Rodrigo P. Silva-Aguiar, Douglas E. Teixeira, Rodrigo A. S. Peres, Diogo B. Peruchetti, Carlos P. Gomes, Alvin H. Schmaier, Patricia R. M. Rocco, Ana Acacia S. Pinheiro, Celso Caruso-Neves

**Affiliations:** 1Carlos Chagas Filho Biophysics Institute, Federal University of Rio de Janeiro, Rio de Janeiro 21941-617, Brazil; 2Clementino Fraga Filho University Hospital, Federal University of Rio de Janeiro, Rio de Janeiro 21941-617, Brazil; 3School of Medicine and Surgery, Federal University of the State of Rio de Janeiro, Rio de Janeiro 21941-617, Brazil; 4Department of Medicine, School of Medicine, Case Western Reserve University, Cleveland, OH 44106, USA; 5University Hospitals Cleveland Medical Center, Cleveland, OH 44106, USA; 6National Institute of Science and Technology for Regenerative Medicine, Rio de Janeiro 21941-902, Brazil; 7Rio de Janeiro Innovation Network in Nanosystems for Health-NanoSAÚDE, Fundação Carlos Chagas Filho de Amparo à Pesquisa do Estado do Rio de Janeiro (FAPERJ), Rio de Janeiro 21045-900, Brazil

**Keywords:** COVID-19, kidney disease, renal protein reabsorption, proximal tubule, megalin, renin-angiotensin system

## Abstract

Since the outbreak of COVID-19 disease, a bidirectional interaction between kidney disease and the progression of COVID-19 has been demonstrated. Kidney disease is an independent risk factor for mortality of patients with COVID-19 as well as severe acute respiratory syndrome coronavirus 2 (SARS-CoV-2) infection leading to the development of acute kidney injury (AKI) and chronic kidney disease (CKD) in patients with COVID-19. However, the detection of kidney damage in patients with COVID-19 may not occur until an advanced stage based on the current clinical blood and urinary examinations. Some studies have pointed out the development of subclinical acute kidney injury (subAKI) syndrome with COVID-19. This syndrome is characterized by significant tubule interstitial injury without changes in the estimated glomerular filtration rate. Despite the complexity of the mechanism(s) underlying the development of subAKI, the involvement of changes in the protein endocytosis machinery in proximal tubule (PT) epithelial cells (PTECs) has been proposed. This paper focuses on the data relating to subAKI and COVID-19 and the role of PTECs and their protein endocytosis machinery in its pathogenesis.

## 1. Introduction

COVID-19, a syndrome caused by severe acute respiratory syndrome (SARS) coronavirus 2 (SARS-CoV-2), has had a major impact on humanity on a global scale since December 2019 [1,2,3]. Initially described as causing SARS associated with high mortality and morbidity, it is now accepted that COVID-19 promotes multiple organ dysfunction [4,5]. Unprecedented scientific effort has advanced our understanding of the disease pathogenesis and progression, leading to significant improvements in the clinical management of COVID-19 such as treatment with antivirals and monoclonal antibodies, and the development of standardized protocols for the treatment of hospitalized patients as well as vaccination campaigns [6]. Despite these advances, knowledge regarding the pathogenesis of multiple organ dysfunction in COVID-19 is still poorly known.

The kidney is a key target of SARS-CoV-2 [7,8,9]. Several reports indicate a high prevalence of acute kidney injury (AKI) in patients with severe COVID-19 [10,11,12]. Conversely, AKI and chronic kidney disease (CKD) are independent risk factors associated with COVID-19 severity and mortality [13,14,15]. According to the Kidney Disease Improving Global Outcomes (KDIGO) guidelines, AKI is characterized by an increase in the serum creatinine level >0.3 mg/dL within 48 h, a 1.5-fold increase in serum creatinine level compared with baseline within 7 days, and/or urine output <0.5 mL/kg/h for 6 h [16]. An important pitfall, however, is the threshold of serum creatinine sensitivity, which increases in response to a decline in the estimated glomerular filtration rate (eGFR) >50% due to renal functional reserve [17]. This process delays the identification of kidney injury, which could be associated with a worse outcome in patients with COVID-19.

The use of biomarkers of tubular injury to determine early kidney damage created a new concept called subclinical AKI (subAKI) [17,18]. So far, there is no consensus regarding a specific definition of subAKI. Based on recent reports, we propose that subAKI represents a large spectrum of parenchymal kidney damage without changes in glomerular function, defined by the KDIGO criteria, associated with the presence of biomarkers of kidney damage in urine [17,18,19]. Tubular injury biomarkers in urine can include β2-microglobulin and kidney injury molecule 1 (KIM-1), markers of proximal injury, and neutrophil gelatinase-associated lipocalin (NGAL), a marker of distal injury [19]. The presence of these biomarkers in the urine is usually associated with microalbuminuria [17,18,19]. At present, subAKI is recognized as an emerging syndrome and a risk factor for the development of AKI and CKD [17,18]. In addition, subAKI predicts adverse outcomes such as the requirement for dialysis and mortality in patients without established AKI [17,20]. Therefore, precocious identification of subAKI could improve the treatment and outcome of patients with kidney injury. However, clinical data regarding the development of subAKI and its involvement with COVID-19 prognosis are still poorly known. In this review, we discuss new perspectives on the possible development of subAKI in patients with COVID-19 and its possible harmful consequences.

## 2. COVID-19 and Kidney Disease

Data from the Acute Disease Quality Initiative Workgroup indicate that approximately 20% of patients hospitalized with COVID-19 develop AKI, and the prevalence increases to approximately 50% in patients admitted to intensive care units [8]. A systematic review comprising 30,639 patients hospitalized with COVID-19 reported similar results [11]. Further, it was observed that the use of kidney replacement therapy ranged from 9% in hospitalized patients to 20% in patients in an intensive care unit.

Conversely, renal dysfunction is an independent risk factor for poor prognosis of patients with COVID-19 [13,14,15]. A study using the OpenSAFELY platform with records from approximately 17 million patients showed that reduced renal function, measured by eGFR, is a key risk factor for COVID-associated mortality [15]. Dialysis or kidney failure increases the risk of mortality due to COVID-19 by 3.7-fold. Patients with COVID-19 who develop AKI have a lower eGFR than patients with COVID-19-independent AKI. These observations corroborate the idea that patients with COVID-19 develop more severe kidney injury [10,11,12].

An increasing number of reports suggest that patients who have had acute COVID-19 might experience persistent renal dysfunction after discharge [14,21,22,23]. Yende and Parikh [21] proposed that subclinical inflammation and injury may persist for many months after the diagnosis of COVID-19, which increases the risk of the development of AKI and CKD. In accordance, Al-Aly et al. [22], in a cohort study (with 6 months follow-up) using the US Department of Veterans Health Affairs database, observed that patients with COVID-19 presented a higher risk of kidney injury even after the first 30 days since diagnosis. In another cohort study on 89,216 patients, it was shown that COVID-19 survivors have an increased risk of developing AKI, a decline in eGFR, progression to ESRD and major kidney disease events [23]. Despite the evidence suggesting that a decline in renal function might be a symptom post-COVID, future clinical studies should verify if persistent renal injury after acute COVID-19 is caused by SARS-CoV-2 infection and/or a result of healthcare restrictions during the pandemic [24].

In a 6-month follow-up study, Huang et al. [25] showed that 13% of patients with COVID-19 without clinical AKI presented a decline in eGFR. This observation indicates the development of a silent kidney injury, such as subAKI, during acute COVID-19. In a retrospective cohort study, markers of subAKI, such as proteinuria and urinary β2-microglobulin, were associated with the severity of COVID-19 and a lower rate of hospital discharge, despite unchanged eGFR [26]. In a recent review, Legrand et al. [11] pointed out the possible role of subAKI in patients with COVID-19. They commented that a late diagnosis, based on the KDIGO guidelines, could contribute to a poor prognosis in patients with COVID-19. Although the development of AKI is usually associated with severe COVID-19, it has been shown that subAKI develops in patients with mild to moderate COVID-19, including children [27,28]. These observations emphasize the importance of the timely identification of subAKI in patients with COVID-19. In a prospective study, the presence of urinary biomarkers was associated with adverse kidney outcomes in patients hospitalized with COVID-19 [29].

## 3. Proximal Tubule Epithelial Cells Are a Target for the Development of subAKI in Patients with COVID-19

The hallmark of subAKI is PT injury associated with a pro-inflammatory and pro-fibrotic phenotype leading to tubule interstitial injury [17,18,19]. Interestingly, PTECs have been proposed to be a primary site of SARS-CoV-2 replication in the kidney [10,30]. Rahmani et al. [31], using scRNA sequencing, showed that PTECs have high potential co-expression of SARS-CoV-2 receptors and proteases involved in cell infection, such as ACE2, NPR-1, TMPRSS2, CTSB, and FURIN.

Alternative routes for SARS-CoV-2 cell infection have also been proposed. Wang et al. [32] showed that CD147 mediates the entry of SARS-CoV-2 in VeroE6 cells (monkey renal cells) and BEAS-2B cells (human bronchiolar cells). The authors revealed that human CD147 (also known as Basigin) allows entry of the virus into non-susceptible BHK-21 cells (hamster renal fibroblast cells). CD147 is highly expressed in the basolateral side of PTECs [33]. Mori et al. [34] showed that LLC-PK1 cells (porcine PTEC line) expressed human KIM-1, which supported the uptake of pseudovirus displaying the SARS-CoV-2 spike protein. Interestingly, KIM-1 is increased in PTECs during the development of subAKI and AKI [35,36].

Based on these observations, it is plausible to postulate that the direct infection of PT epithelial cells (PTECs) with SARS-CoV-2 could be a central mechanism for the development of subAKI in patients with COVID-19. In agreement, Caceres et al. [37] showed that there is a correlation between the presence of SARS-CoV-2 in the urine and the incidence of AKI and mortality in patients with COVID-19. However, so far, there are no studies showing a direct correlation between SARS-CoV-2 infection in renal cells and the development of AKI in patients with COVID-19. Multiple organ damage could also be involved in the deleterious effect of SARS-CoV-2 on kidney function as seen with hemodynamic changes and/or an exacerbated immune response, such as cytokine storm [1,2,4,38]. PTECs express different cytokine receptors, including IL-6R [39], IL-4R [40], and TNF-αR [41]. There is an association between plasma levels of IL-6 and the development of AKI in patients with COVID-19 [42]. Medeiros et al. [43] showed that AKI associated with COVID-19 is accompanied by significant alterations in circulating levels of immune mediators such as IFN-γ, IL-2, IL-6, TNF-α, IL-1Ra, IL-10, and VEGF. They postulated that this phenomenon could contribute to the establishment of AKI. In another study, using urine collected from 29 patients with COVID-19 admitted to the intensive care unit, strong correlations between pro-inflammatory cytokines and AKI were observed [44].

Saygili et al. [28] showed an association between the neutrophil count and the development of AKI, pointing out the role of inflammation in this process. This hypothesis is reinforced by the observation, in a comparative study, that the prevalence of AKI in patients with COVID-19 is similar to that in patients infected with seasonal influenza [45]. Despite the similar prevalence, COVID-19 was associated with a higher risk of developing stage 3 AKI and higher proteinuria rates than influenza infection, suggesting that additional mechanisms participate in the renal injury associated with COVID-19.

## 4. Role of Proximal Tubule Albumin Reabsorption in the Development of subAKI

It has been suggested that PTEC megalin-mediated protein reabsorption may initiate the tubule interstitial injury observed in subAKI [46,47,48]. Proteinuria is an independent risk factor for the severity of COVID-19, the development of AKI, and the progression of CKD [49,50,51]. The prevalence of proteinuria was found to be high among patients with COVID-19, even those who did not develop AKI [52]. Furthermore, the correlation between proteinuria and glycosuria with the severity of the COVID-19 disease has been demonstrated [53,54]. These data indicate a role for PT protein reabsorption in this process. In this context, we focused on the possible role of megalin, a PTEC protein receptor, as a sensor and integrator between the development of tubular proteinuria and subAKI in patients with COVID-19.

Tubular proteinuria is associated with modifications in protein reabsorption in PTECs, which occurs by a receptor-mediated endocytosis mechanism [55,56]. The receptor is formed by the association of three proteins: megalin, cubilin and amnionless. Megalin (also as known LDL-related protein 2 [LRP2]) is an LDL-like family member highly expressed in the luminal membrane of PTECs and has a critical role in the internalization of protein/receptor complex [56,57]. Megalin is considered a molecular platform integrating extracellular and intracellular signals and working as a sensor for changes in tubular albumin concentration as well as a target for different signals. It works as a scaffold protein, anchoring different signaling and structural molecules such as protein kinase B (Akt) [58], disable homolog 2 (Dab2) [59], and autosomal recessive hypercholesterinemia (ARH) [60]. It is also a target for post-translational modifications, such as phosphorylation, O-GlcNAcylation and proteolysis [61,62,63].

Megalin has been shown to be involved in the survival and death of PTECs, regulation of the immune response, and PTEC metabolism [58,64,65,66]. Long et al. [65] showed that megalin knockout (KO) modulates the expression of PI3K/Akt, genes linked to metabolism and pro-inflammatory response and signaling pathways. The association between tubule interstitial injury and megalin was observed in lrp2−/− mice [48,66], which present some characteristics of PTEC dysfunction such as Fanconi syndromes, albuminuria, phosphaturia, glycosuria, and markers of tubular injury such as KIM-1 [44]. Similar characteristics are observed in patients with Donnai–Barrow/Facio-Oculo-Acoustico-Renal (DB/FOAR) syndrome, caused by megalin loss-of-function mutations [67,68]. Megalin is also the antigen target of human anti-brush border antibody kidney disease (ABBA disease), which causes primary renal tubule interstitial disease [69]. Megalin deficiency progresses to glomerular dysfunction, suggesting that the tubular injury could precede glomerular dysfunction [48].

Kormann et al. [70] showed a high prevalence of Fanconi-like syndrome in a retrospective cohort of 42 patients with laboratory-confirmed COVID-19 without a history of kidney disease. The authors observed at least two PT abnormalities (incomplete Fanconi syndrome) in 75% of the patients. The main disorders were proteinuria (88% of the patients), renal phosphaturia (55% of the patients), hyperuricosuria (43% of the patients), and normoglycemic glycosuria (30% of the patients). Werion et al. [71] showed that patients with COVID-19 present low molecular weight proteinuria and increased urinary excretion of β2-microglobulin associated with reduced expression of megalin and focal proximal tubular necrosis. In agreement, proteomic analysis of urine samples from patients with COVID-19 revealed a decrease in megalin excretion. The authors suggested a possible correlation with the decrease in PT protein reabsorption observed in patients with COVID-19 [72]. Furthermore, in a recent work, our group showed that spike protein decreases megalin expression and albumin endocytosis in a model of PTECs [73]. Importantly, these effects were not associated with changes in aquaporin 1, indicating specificity for the albumin endocytosis machinery.

Menez et al. [29] showed that the increase in NGAL and KIM-1, biomarkers of tubular injury, is correlated with adverse kidney outcomes in patients hospitalized with COVID-19. They also mention that there was evidence for the development of subAKI based on increased levels of these biomarkers without the establishment of AKI. Vogel et al. [74] showed that Kim-1 could be predictive of AKI at an early stage. Furthermore, increased KIM-1 was significantly correlated with admission to the intensive care unit in contrast to the serum creatinine level. These observations could explain the presence of albuminuria and urinary tubular injury markers in patients with COVID-19 before changes in eGFR can be observed [23,26].

Furthermore, in a recent work, Canney et al. [75] showed that patients with previous glomerular disease have an increased risk of relapse when exposed to a second or third dose of the COVID-19 vaccine, measured by the increase in proteinuria and/or the decline of glomerular function. Since vaccines are based on the spike protein, these results indicate that spike induces a cellular response in susceptible patients’ subgroups. However, these results must be interpreted with care once the authors mentioned that there is a low absolute increased risk of relapse. Additionally, the authors did not mention if the kidney damage was transient or not. Further studies should be performed to confirm other possible adverse side effects in kidney disease patients.

## 5. Megalin and the Renin-Angiotensin System: Possible Link between Kidney Injury and Severe COVID-19

The renin–angiotensin system (RAS) is formed by a complex network of peptides and proteases with an important role in the control of the internal milieu [76,77,78]. Two different types of RAS have been described: systemic and tissular. Despite the same structure, they are activated by distinct signals and have different roles under physiologic and pathophysiologic conditions [76,77,78,79]. RAS involves two main axes: (1) angiotensin II (Ang II)/AT1 receptor (AT1R) and (2) angiotensin-(1-7) (Ang-(1-7)/Mas receptor (MASR) and Ang II/AT2 receptor (AT2R). Activation of the Ang II/AT1R axis induces vasoconstriction and pro-inflammatory and pro-fibrotic effects [76,80,81,82], whereas activation of Ang-(1-7)/MASR and/or Ang II/AT2R offsets these effects [77,83,84,85]. In this regard, angiotensin-converting enzyme type 2 (ACE2), which breaks down angiotensin II (Ang II) into Ang-(1-7), has a crucial role in the balance of the effects triggered by both pathways [77]. Deletion of ACE2 is associated with an increase in the intrarenal Ang II level and the development of kidney injury [86,87,88].

One attractive hypothesis is that SARS-CoV-2 cells promote the overactivation of the Ang II/(AT1R) axis as a consequence of ACE2 inhibition [89,90,91]. This inhibition involves spike protein-mediated binding of SARS-CoV-2 to ACE2, leading to its cleavage and internalization [1]. Liu et al. [92] showed an increase in the plasma Ang II level in Chinese patients with COVID-19 in 2019, which was associated with viral load and lung injury. Furthermore, Zoufaly et al. [93], in a case report, showed that the treatment of a 45-year-old woman diagnosed with severe COVID-19 with human recombinant soluble ACE2 (hrsACE2) for 9 days decreased the Ang II level, pro-inflammatory interleukins such as IL-6 and IL-8, inflammation marker ferritin, and C-reactive protein. These effects were followed by the recovery of the patient. In agreement, the severity of COVID-19 was associated with increased plasma Ang II levels [94].

On the other hand, some studies found no changes or even a decrease in plasma Ang II levels in patients with COVID-19 [95,96]. However, the level of Ang II in specific tissues was not assessed. Recently, Ensor et al. [97] observed that injection of spike protein in male mice increased the Ang II level in the lungs and inhibited ACE2 activity in adipocytes. These results suggest that tissue RAS may have an important role during the development of kidney damage in patients with COVID-19 rather than systemic RAS. However, how renal RAS is correlated with the development of kidney damage in patients with COVID-19 is still an open matter.

It has been proposed that the cortical Ang II/AT1R axis has a role in the development of subAKI [47,98]. The activation of this axis decreases megalin expression and albumin endocytosis and induces the development of pro-inflammatory and pro-fibrotic phenotypes contributing to the genesis of tubule interstitial injury [47]. Based on these observations, we postulate that overactivation of the Ang II/AT1R axis induced by ACE2 inhibition and its inhibitory effect on megalin expression could be behind the tubular proteinuria and development of subAKI observed in patients with COVID-19.

## 6. Megalin Expression as a Risk Factor for the Development of Severe Kidney Injury in Patients with COVID-19 and Chronic Degenerative Disease

Chronic degenerative diseases such as hypertension and diabetes are strictly correlated to the development of kidney disease [99,100,101,102], and they are risk factors for higher mortality and morbidity among patients with COVID-19 [13,14,15,103]. Thakur et al. [14], using a meta-analysis of 120 studies with 125,446 patients, observed that hypertension, obesity, and diabetes are the most prevalent comorbidities, and CKD or other renal diseases are the highest risk factors for the severity of COVID-19. In addition, the authors observed that patients with kidney disease have a higher mortality risk for COVID-19 even compared with those with respiratory disease. Cai et al. [103], using a meta-analysis, showed that patients with diabetes and hypertension have a higher probability of developing AKI when infected with COVID-19.

It has been shown that the early stage of kidney injury induced by essential hypertension or diabetes involves alterations in PTEC megalin-mediated protein reabsorption, which precedes changes in the eGFR [62,104,105,106,107]. In the early stage of diabetes, tubular albuminuria was reported without changes in glomerular albumin permeability and eGFR [104,105]. The mechanism underlying this process is likely to involve an increase in the glucose concentration in the lumen of PTs and its influx through sodium-glucose cotransporters (SGLTs), leading to decreased megalin expression [106]. Megalin expression is also decreased in patients with diabetes [107]. This observation is consistent with experimental and clinical data showing the antiproteinuric effect of SGLT inhibitors in patients with and without diabetes [108,109,110].

As previously observed with diabetes, young spontaneously hypertensive rats (SHR) developed proteinuria before the establishment of hypertension [111]. Furthermore, luminal membrane megalin expression in PTECs of SHRs was decreased and was associated with tubular proteinuria [62]. These results corroborate the clinical evidence that albuminuria precedes the establishment of essential hypertension [112,113,114]. In a follow-up study with 2430 patients, a higher blood pressure trajectory in childhood correlates with higher albuminuria and risk of development of subAKI in adulthood [115].

How is renal megalin expression modulated in hypertension and diabetes? One important clue comes from the observation that the renal Ang II/AT1R axis is involved in kidney injury induced by hypertension and diabetes [116,117]. It was shown that the AT1R antagonist losartan abolished the decrease in PTEC megalin expression and albuminuria observed in SHRs and a diabetic animal model [111,118,119]. In addition, a decrease in ACE2 expression in the kidney was observed in hypertension-induced nephropathy, indicating overactivation of the Ang II/AT1R axis [120]. In agreement, SHRs present an increase in the Ang II level in the renal cortex but not in plasma [121].

Based on these observations, we can postulate that previous modifications in megalin-mediated albumin endocytosis and in the Ang II/AT1R axis in PTECs, i.e, in diabetes and hypertension, are risk factors for the development of severe kidney injury in COVID-19.

## 7. Conclusions and Perspective

The current literature shows a clear correlation between kidney injury and the severity of COVID-19 with PTECs a target in this process [7,8,9]. The mechanism underlying the development of tubular injury likely involves a complex network leading to a pro-inflammatory and pro-fibrotic phenotype. We postulate that megalin, a component of albumin endocytosis machinery, works as a sensor in the development of renal tubular injury and tubular proteinuria observed in patients with COVID-19.

We proposed that PT epithelial damage, observed in patients with COVID-19, involves a decrease in ACE2 activity, leading to an increase in the level of Ang II and consequent activation of the Ang II/AT1R axis (Figure 1). Once activated, the Ang II/AT1R axis promotes a decrease in megalin expression and albumin endocytosis leading to tubular albuminuria as observed in patients with COVID-19 [71]. Tubule interstitial injury, characterized by a pro-inflammatory and pro-fibrotic phenotype, may arise by modifications in the albumin endocytosis machinery and/or due to direct modulation of the immune response mediated by the Ang II/AT1R axis. Together, these factors promote the genesis of subAKI in acute COVID-19.

If the repair mechanism works and the insult ceases, there is no progression from subAKI to AKI (Figure 2A,B, blue dotted line). In this case, a “fingerprint” is established, increasing epithelial sensitivity to a new renal insult. This process correlates to a higher risk of patients with COVID-19 developing AKI and/or CKD (Figure 2A,B, blue dotted line). On the other hand, if the repair mechanisms fail, there could be progression to AKI, associated with a decrease in eGFR, as observed in patients with severe COVID-19 (Figure 2A,B, blue solid line). In patients with comorbidities such as hypertension and diabetes where kidney injury and renal Ang II/AT1R axis activation are already established, an increase in the sensitivity of the system can be observed (Figure 2A,B, red solid line). Therefore, a new insult during acute COVID-19 could worsen the existing renal damage, leading to a faster progression of kidney disease and the development of severe COVID-19 associated with increased mortality.

Based on the discussion, we propose that the determination of proximal tubule damage biomarkers should be introduced in the clinical routine for patients with symptomatic COVID-19, especially those who require hospitalization or have comorbidities such as hypertension and diabetes. This process could be assessed by monitoring urinary β2-microglobulin and KIM-1 excretion associated with microalbuminuria. These biomarkers are currently being monitored in specific clinical analyses [122,123]. This approach could identify the early stage of subAKI, allowing timely treatment and avoiding progression to kidney damage.

Angiotensin II receptor blockers (ARBs) are used as a frontline therapy to halt the progression of renal disease associated with a decrease in proteinuria independently of blood pressure in patients who do not have COVID-19 [124,125]. In an open-label randomized trial with 158 patients, Duarte et al. [126] showed that treatment with telmisartan (80 mg/twice a day) for 14 days decreased the level of C-reactive protein, time to discharge, and 30-day mortality. On the other hand, in a double-blind randomized study with 117 patients with symptomatic COVID-19, Puskarich et al. [127] did not observe any modifications in hospitalization within 15 days and the viral load treated with losartan (25 mg/twice a day) for 10 days. To our knowledge, there are no studies reporting the effect of ARBs on the kidney injury observed in patients with symptomatic COVID-19, and this hypothesis should be tested to confirm a possible beneficial effect.

## Figures and Tables

**Figure 1 ijms-23-14193-f001:**
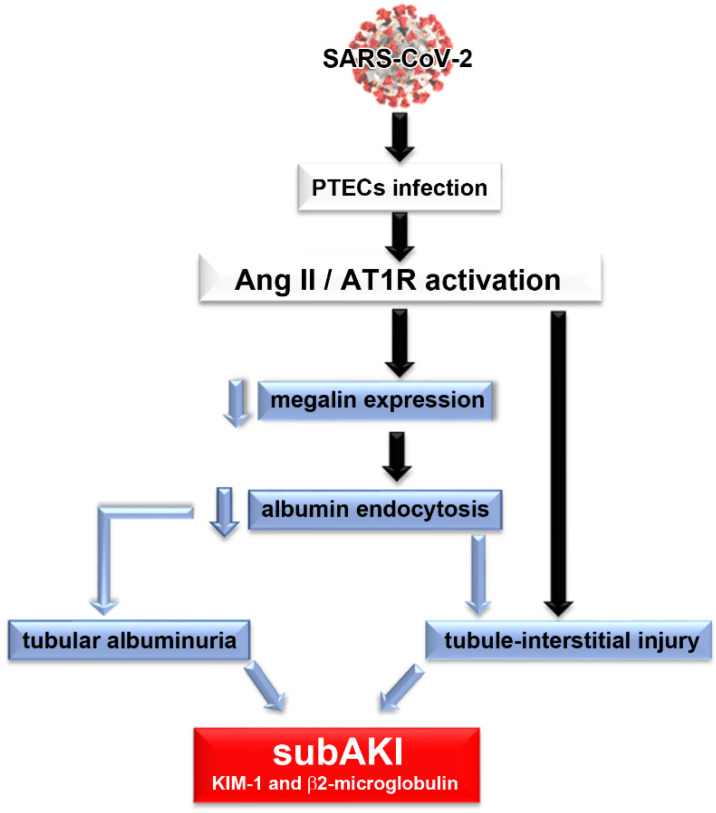
Proposed model of the genesis of subclinical acute kidney injury in patients with COVID-19. We proposed that PTEC SARS-CoV-2 infection promotes overactivation of the Ang II/AT1R axis, which leads to a decrease in megalin expression, a decrease in albumin endocytosis and consequently, tubular albuminuria. Development of tubule interstitial injury could be caused by changes in the albumin endocytosis machinery per se and/or due to a direct effect of overactivation of the Ang II/AT1R axis. All these conditions characterize the development of subAKI. In this case, tubular injury biomarkers, such as KIM-1 and β2-microglobulin, and albuminuria could be observed in the urine of patients.

**Figure 2 ijms-23-14193-f002:**
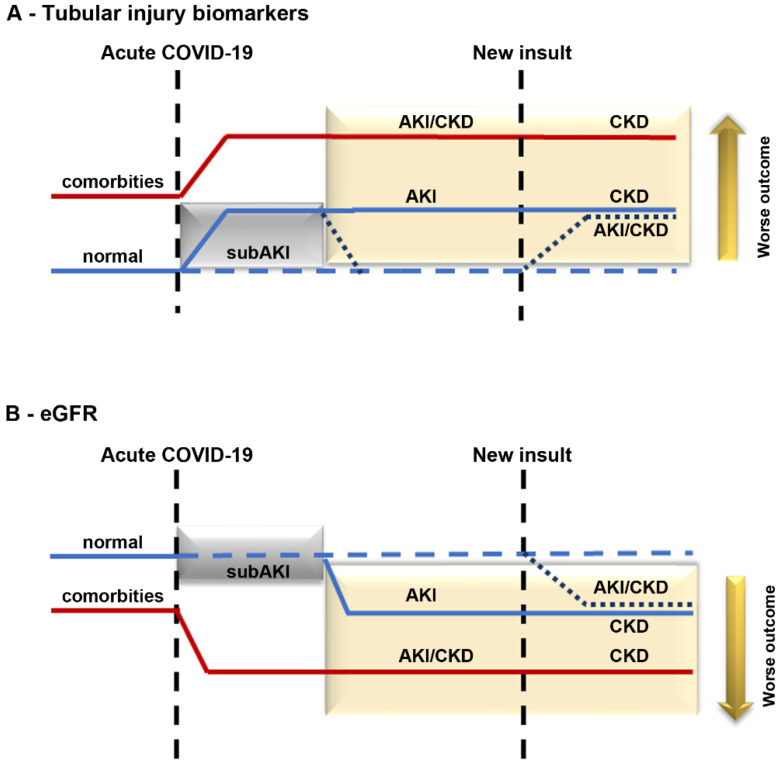
Proposed model for progression of kidney disease in patients with COVID-19. Kidney injury in patients with COVID-19 without (blue line) or with (red line) comorbidities. In an acute phase, patients with COVID-19 develop subAKI (gray box) characterized by an increase in tubular injury biomarkers (**A**) without changes in eGFR (**B**). If the repair mechanism works, the levels of tubular injury biomarkers return to normal (dotted blue line). On the other hand, subAKI can progress to AKI if the repair mechanism is insufficient (solid blue line inside the yellow box). In the case of patients with already established comorbidities, such as hypertension and diabetes (red solid line), an increase in the level of tubular injury biomarkers and a decrease in eGFR can be observed. If these patients are infected with COVID-19, their renal injury is aggravated, leading to faster progression of kidney injury. Patients with COVID-19 who develop any kind of renal injury, including subAKI (new insult), are at higher risk of developing kidney disease post-COVID-19.

## Data Availability

Not applicable.

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
