# Peer review of "Subclinical Acute Kidney Injury in COVID-19: Possible Mechanisms and Future Perspectives"

_ijms, 2022, doi:10.3390/ijms232214193_

Round 1
Reviewer 1 Report
I read with interest the review article from Silva-Aguiar and colleagues. In this manuscript, the authors aimed to discuss the relevance of mild kidney function alterations in COVID-19 prior to the development of AKI, which are referred as “subclinical AKI”. In this context, they explored the role megalin-mediated protein reabsorption by proximal tubular epithelial cells, which could be altered in response to SARS-CoV-2 infection. Since kidney injury is one of the main complications observed in the ICU for COVID-19 patients, and directly impacts on mortality, this review is relevant for the scientific community. Please find my considerations below.
Major comments:
- What is the definition of subAKI? The authors mentioned that measurements such as proteinuria and β2M can be used; however, I strongly suggest that the authors describe in detail which parameters are included (or suggested) for the diagnosis of this condition.
- Lines 128-129: The authors mentioned that there are no studies showing that renal cell infection by SARS-CoV-2 can directly lead to AKI development. Nevertheless, I suggest to discuss the evidence that we have so far. For example, a recent study showed that increased urinary SARS-CoV-2 viral load is associated with AKI and poor outcomes in COVID-19 patients. (Caceres P, et al. JASN, 2021).
- Indeed, several studies have shown that inflammation is pivotal for the development of severe COVID-19 and is also involved in the pathogenesis of renal injury. The authors only mentioned high IL-6 and neutrophilia in AKI patients, which is quite generic. There are an increasing number of evidence showing that these patients also present important alterations of inflammatory mediators, both circulant and urinary. Please see 1) Medeiros et al. Acute kidney injury associated to COVID-19 leads to a strong unbalance of circulant immune mediators. Cytokine. 2022; 2) Laudanski et al. Longitudinal Urinary Biomarkers of Immunological Activation in Covid-19 Patients without Clinically Apparent Kidney Disease versus Acute and Chronic Failure. Sci Rep 2021; 3) Gradin, A. Urinary Cytokines Correlate with Acute Kidney Injury in Critically Ill COVID-19 Patients. Cytokine 2021; and others.
- Lines 146-147: The authors cited the study of Huart et al to show that, even in the absence of AKI, COVID-19 patients present proteinuria. Since proteinuria is one of the measurements suggested for subAKI definition and the authors are focused on the investigation of PTEC-mediated protein reabsorption as a mechanism of tubular injury, I suggest that the authors discuss other studies concerning the analysis of proteinuria in COVID-19 patients. Please see: 1) Erdogan et al. Is There an Association between Urine Biochemical Parameters on Admission and the Severity OF COVID‐19? Int J Clin Pract 2021; 2) Liu et al. The Value of Urine Biochemical Parameters in the Prediction of the Severity of Coronavirus Disease 2019. Clinical Chemistry and Laboratory Medicine (CCLM) 2020; and others.
- KIM-1 is mentioned along the text as an important marker of tubular injury. The authors should include other studies showing KIM-1 alterations in COVID-19.
- Lines 178-179: I suggest to explain the abnormalities observed by Kormann et al. The main finding was proteinuria, in 88% of patients.
- Lines 179-182: Did Werion et al the only group showing reduced megalin expression in COVID-19 patients? Maybe some proteomic-based studies could help to strengthen the hypothesis of reduced megalin expression leading to alterations in protein handling on proximal tubules and subAKI development. The authors should also discuss their recently published paper concerning the effect of SARS-CoV-2 Spike protein on the inhibition of megalin-mediated protein endocytosis.
Minor comments:
- The abbreviations “SARS-CoV-2” (line 32), “PT” (line 56 – also check the position in the sentence), “SGLT” (line 243) and ARB (line 325) should be defined.
- Lines 74-75: Since 2018, KDIGO recommends to use the term “kidney failure” instead of “ESRD” since it may be sensitive to patients.
- Lines 75-78: I believe that the lower eGFR observed at baseline suggests that poor kidney function in the early stages of COVID-19 progression could be considered as a risk factor for AKI development.
- Lines 190-191: The word “physiologic” is repeated.
- I think that Figure 2 was well planned and efficiently reflects the authors hypothesis. Please check the spelling of the word “comorbidities”. I also suggest to change “normal” to “without comorbidities” or “healthy”.
Author Response
Answer to Reviewers
Reviewer 1
First of all, we would like to mention that the manuscript has been sent to English revision by a native speaker (authorserv@gmail.com) as requested by the Reviewer.
Major comments:
Question 1 - What is the definition of subAKI? The authors mentioned that measurements such as proteinuria and β2M can be used; however, I strongly suggest that the authors describe in detail which parameters are included (or suggested) for the diagnosis of this condition.
Answer – So far there is not a consensus regarding a specific definition of subAKI in literature. In introduction section (P. 2, line 53) we mention: “The definition of subAKI includes a large spectrum of parenchymal kidney damage with or without minimal damage in glomerular structure and function even when the KDIGO criteria have not yet been achieved. (PTs) [16-18].”. To clarify this issue this text has been replaced by:
- 2, line 54
“So far, there is no consensus regarding a specific definition of subAKI. Based on recent reports, we propose that subAKI represents a large spectrum of parenchymal kidney damage without changes in glomerular function, defined by the KDIGO criteria, associated with the presence of biomarkers of kidney damage in urine [17-19]. Tubular in-jury biomarkers in urine can include β2-microglobulin and kidney injury molecule 1 (KIM-1), markers of proximal injury, and neutrophil gelatinase-associated lipocalin (NGAL), a marker of distal injury [19]. The presence of these biomarkers in the urine is usually associated with microalbuminuria [17-19].”
Question 2 - Lines 128-129: The authors mentioned that there are no studies showing that renal cell infection by SARS-CoV-2 can directly lead to AKI development. Nevertheless, I suggest to discuss the evidence that we have so far. For example, a recent study showed that increased urinary SARS-CoV-2 viral load is associated with AKI and poor outcomes in COVID-19 patients. (Caceres P, et al. JASN, 2021).
Answer - We agree with reviewer and the following text has been added:
P.3, line 131
“In agreement, Caceres et al. [37] showed that there is a correlation between the presence of SARS-CoV-2 in the urine and the incidence of AKI and mortality in patients with COVID-19.”
Question 3 - Indeed, several studies have shown that inflammation is pivotal for the development of severe COVID-19 and is also involved in the pathogenesis of renal injury. The authors only mentioned high IL-6 and neutrophilia in AKI patients, which is quite generic. There are an increasing number of evidence showing that these patients also present important alterations of inflammatory mediators, both circulant and urinary. Please see 1) Medeiros et al. Acute kidney injury associated to COVID-19 leads to a strong unbalance of circulant immune mediators. Cytokine. 2022; 2) Laudanski et al. Longitudinal Urinary Biomarkers of Immunological Activation in Covid-19 Patients without Clinically Apparent Kidney Disease versus Acute and Chronic Failure. Sci Rep 2021; 3) Gradin, A. Urinary Cytokines Correlate with Acute Kidney Injury in Critically Ill COVID-19 Patients. Cytokine 2021; and others.
Answer – We agree with reviewer and the following text was added:
- 3 line 140
“Medeiros et al. [43] showed that AKI associated with COVID-19 is accompanied by significant alterations in circulating levels of immune mediators such as IFN-γ, IL-2, IL-6, TNF-α, IL-1Ra, IL-10, and VEGF. They postulated that this phenomenon could contrib-ute to the establishment of AKI. In another study, using urine collected from 29 patients with COVID-19 admitted to the intensive care unit, strong correlations between pro-inflammatory cytokines and AKI were observed [44].”
Question 4 - Lines 146-147: The authors cited the study of Huart et al to show that, even in the absence of AKI, COVID-19 patients present proteinuria. Since proteinuria is one of the measurements suggested for subAKI definition and the authors are focused on the investigation of PTEC-mediated protein reabsorption as a mechanism of tubular injury, I suggest that the authors discuss other studies concerning the analysis of proteinuria in COVID-19 patients. Please see: 1) Erdogan et al. Is There an Association between Urine Biochemical Parameters on Admission and the Severity OF COVID‐19? Int J Clin Pract 2021; 2) Liu et al. The Value of Urine Biochemical Parameters in the Prediction of the Severity of Coronavirus Disease 2019. Clinical Chemistry and Laboratory Medicine (CCLM) 2020; and others.
Answer – We agree with Reviewer and the following text:
- 4, line 159
“The prevalence of proteinuria was found to be high among patients with COVID-19, even those who did not develop AKI [48], indicating a role for PT protein reabsorption in this process.
has been replaced by
“The prevalence of proteinuria was found to be high among patients with COVID-19, even those who did not develop AKI [52]. Furthermore, correlation between proteinuria and glycosuria with the severity of the COVID-19 disease has been demonstrated [53,54]. These data indicate a role for PT protein reabsorption in this process.”
Question 5 - KIM-1 is mentioned along the text as an important marker of tubular injury. The authors should include other studies showing KIM-1 alterations in COVID-19.
Answer - To clarify this issue the following text was added:
- 5, line 206
“Menez et al. [74] showed that the increase in NGAL and KIM-1, biomarkers of tubular injury, is correlated with adverse kidney outcomes in patients hospitalized with COVID-19. They also mention that there was evidence of development of subAKI based on increased levels of these biomarkers without establishment of AKI. Vogel et al. [75] showed that Kim-1 could be predictive of AKI at an early stage. Furthermore, in-creased KIM-1 was significantly correlated with admission to the intensive care unit in contrast to the serum creatinine level.”
Question 6 - Lines 178-179: I suggest to explain the abnormalities observed by Kormann et al. The main finding was proteinuria, in 88% of patients.
Answer – To clarify this issue the following text:
P.4, line 193
“The authors observed at least two PT abnormalities (incomplete Fanconi syndrome); al-buminuria was more prevalent.”
has been replaced by
“The authors observed at least two PT abnormalities (incomplete Fanconi syndrome) in 75% of the patients. The main disorders were proteinuria (88% of the patients), renal phosphaturia (55% of the patients), hyperuricosuria (43% of the patients), and normo-glycemic glycosuria (30% of the patients).”
Question 7 - Lines 179-182: Did Werion et al the only group showing reduced megalin expression in COVID-19 patients? Maybe some proteomic-based studies could help to strengthen the hypothesis of reduced megalin expression leading to alterations in protein handling on proximal tubules and subAKI development. The authors should also discuss their recently published paper concerning the effect of SARS-CoV-2 Spike protein on the inhibition of megalin-mediated protein endocytosis.
Answer – We agree with reviewer and the following text was added:
P.4, line 199
“In agreement, proteomic analysis of urine samples from patients with COVID-19 revealed a decrease in megalin excretion. The authors suggested a possible correlation with the decrease in PT protein reabsorption observed in patients with COVID-19 [72]. Furthermore, in a recent work, our group showed that spike protein decreases megalin expression and albumin endocytosis in a model of PTECs [73]. Importantly, these ef-fects were not associated with changes in aquaporin 1, indicating specificity for the al-bumin endocytosis machinery.”
Minor comments:
Question 1 - The abbreviations “SARS-CoV-2” (line 32), “PT” (line 56 – also check the position in the sentence), “SGLT” (line 243) and ARB (line 325) should be defined.
Answer – We agree, and definitions were incorporated in the text.
Question 2 - Lines 74-75: Since 2018, KDIGO recommends to use the term “kidney failure” instead of “ESRD” since it may be sensitive to patients.
Answer - We agree, and the term has been incorporated.
Question 3 - Lines 75-78: I believe that the lower eGFR observed at baseline suggests that poor kidney function in the early stages of COVID-19 progression could be considered as a risk factor for AKI development.
Answer – We agree with the idea of the Reviewer. Although, our citation in the text means that patients with AKI associated COVID-19 presents lower eGFR than AKI patients without COVID-19. This observation corroborates the idea that patients with COVID-19 develop more severe kidney injury. To clarify this issue the following text:
P.2, line 81
“Patients with COVID-19 who develop AKI have a lower eGFR at baseline than patients with COVID-19-independent AKI.”
has been replaced by
“Patients with COVID-19 who develop AKI have a lower eGFR than patients with COVID-19-independent AKI.”
Question 4 - Lines 190-191: The word “physiologic” is repeated.
Answer – We mentioned “… physiologic and pathophysiologic conditions…”.
Question 5 - I think that Figure 2 was well planned and efficiently reflects the authors hypothesis. Please check the spelling of the word “comorbidities”. I also suggest to change “normal” to “without comorbidities” or “healthy”.
Answer – We agree, and modifications have been done.
Reviewer 2 Report
Major Comments
This is a general review of AKI in COVID-19 illness requiring hospitalization.
Major Comment
Please insert a paragraph on iatrogenic contributions to AKI due to Remesivir. Please cite the Nov 2020 warning that Remdesivir should not be used in hospitalized patients. Please cite what percent of patients with COVID-19 receive Remdesivir.
Please include Remdesivir contributing to AKI in Figures 1 and 2.
Please insert a paragraph of COVID-19 hospitalization after COVID-19 vaccination breakthrough. Include mention of vaccine-induced Spike protein expression and damage in the kidney as a predisposing factor to AKI once hospitalized. Cite the literature on AKI due to COVID-19 vaccination.
Since the majority of those hospitalized are among COVID-19 vaccine recipients when fairly recorded (Canada, UK, EU, SA, and Israel) please be sure to mention that AKI among hospitalized patients in the setting of COVID-19 must always consider systemic Spike protein loading and inflammation with antecedant COVID-19 vaccination.
Author Response
Reviewer 2
Major Comments
Question 1 - This is a general review of AKI in COVID-19 illness requiring hospitalization.
Answer – We think there are some mistakes of the Reviewer because this is not a general review about AKI in COVID-19 illness. As described in the title and along of the text, this review is about subclinical Acute Kidney Injury (subAKI) and the possible mechanisms involved in its pathogenesis as mentioned in the abstract (P.1, line 27): “Presently, we focus on the data relating to subAKI and COVID-19 and the role of PTECs and their protein endocytosis machinery in its pathogenesis.”. This topic has been highlighted in the last decade in nephrology field as you can observe in several important reviews published in high quality journals (1-5).
Question 2 - Please insert a paragraph on iatrogenic contributions to AKI due to Remesivir. Please cite the Nov 2020 warning that Remdesivir should not be used in hospitalized patients. Please cite what percent of patients with COVID-19 receive Remdesivir.
Question 3 - Please include Remdesivir contributing to AKI in Figures 1 and 2.
Answer 2 and 3 - We think that the treatment of COVID-19 patients with Remdesivir is an important topic, but it is not in scope of the present review.
Question 4 - Please insert a paragraph of COVID-19 hospitalization after COVID-19 vaccination breakthrough. Include mention of vaccine-induced Spike protein expression and damage in the kidney as a predisposing factor to AKI once hospitalized. Cite the literature on AKI due to COVID-19 vaccination.
Question 5 - Since the majority of those hospitalized are among COVID-19 vaccine recipients when fairly recorded (Canada, UK, EU, SA, and Israel) please be sure to mention that AKI among hospitalized patients in the setting of COVID-19 must always consider systemic Spike protein loading and inflammation with antecedant COVID-19 vaccination.
Answer 4 and 5 – Again, we think the Reviewer did not understand the goal of the present Review. These topics mentioned by the Reviewer could be found in other works or reviews available in pubmed database. But they are not the goal of the present review as mentioned in the abstract (P.1, line 27): “Presently, we focus on the data relating to subAKI and COVID-19 and the role of PTECs and their protein endocytosis machinery in its pathogenesis.”
References:
1- Haase M, Kellum JA, Ronco C. Subclinical AKI--an emerging syndrome with important consequences. Nat Rev Nephrol. 2012 Dec;8(12):735-9. doi: 10.1038/nrneph.2012.197.
2- Zou C, Wang C, Lu L. Advances in the study of subclinical AKI biomarkers. Front Physiol. 2022 Aug 24;13:960059. doi: 10.3389/fphys.2022.960059.
3- Vanmassenhove J, Van Biesen W, Vanholder R, Lameire N. Subclinical AKI: ready for primetime in clinical practice? J Nephrol. 2019 Feb;32(1):9-16. doi: 10.1007/s40620-018-00566-y.
4- Ostermann M, Zarbock A, Goldstein S, Kashani K, Macedo E, Murugan R, Bell M, Forni L, Guzzi L, Joannidis M, Kane-Gill SL, Legrand M, Mehta R, Murray PT, Pickkers P, Plebani M, Prowle J, Ricci Z, Rimmelé T, Rosner M, Shaw AD, Kellum JA, Ronco C. Recommendations on Acute Kidney Injury Biomarkers From the Acute Disease Quality Initiative Consensus Conference: A Consensus Statement. JAMA Netw Open. 2020 Oct 1;3(10):e2019209. doi: 10.1001/jamanetworkopen.2020.19209. Erratum in: JAMA Netw Open. 2020 Nov 2;3(11):e2029182.
5- Kellum JA, Romagnani P, Ashuntantang G, Ronco C, Zarbock A, Anders HJ. Acute kidney injury. Nat Rev Dis Primers. 2021 Jul 15;7(1):52. doi: 10.1038/s41572-021-00284-z.
Round 2
Reviewer 1 Report
The authors have satisfactorily addressed all the concerns.
Author Response
We appreciate the reviewer comment.
Reviewer 2 Report
Minor Comments
Ln 39 change to read: "...prevention of mortality with early combination medical regimens."
There are no RCT's of vaccines demonstrating reductions in mortality and there are no valid observational studies demonstrating this theoretical basis
Discussion, please mention that the advent of COVID-19 vaccines that introduce large quantities of systemic Spike protein are likely to intensifiy the mechanisms discussed in this paper resulting in even greater progression of CKD as shown by Canney et al.
Canney reference: Canney M, Atiquzzaman M, Cunningham AM, Zheng Y, Er L, Hawken S, Zhao Y, Barbour SJ. A Population-Based Analysis of the Risk of Glomerular Disease Relapse after COVID-19 Vaccination. J Am Soc Nephrol. 2022 Nov 4:ASN.2022030258. doi: 10.1681/ASN.2022030258. Epub ahead of print. PMID: 36332971.
Author Response
Answer to Reviewer 2
Question 1: Ln 39 change to read: "...prevention of mortality with early combination medical regimens."
Answer: We understand that the Reviewer is talking about the treatment with antiviral and monoclonal antibodies, that has been recently approved, for the treatment of nonhospitalized adults with COVID-19. To clarify this issue the text:
- 1, line 39
“…as well as prevention of mortality with vaccines and specific antivirals…”
has been replaced by
“…such as treatment with antivirals and monoclonal antibodies, and the development of standardized protocols to the treatment of hospitalized patients as well as vaccination campaigns …”
Question 2: There are no RCT's of vaccines demonstrating reductions in mortality and there are no valid observational studies demonstrating this theoretical basis.
Answer: The efficacy of the vaccines used has been shown in different works as can be observed in the references listed below. In fact, to be approved by WHO and national healthy agencies, the vaccine must be tested in a RCT [1-4]. Nowadays, all epidemiologic results obtained have reinforced the importance of vaccination campaign to handling COVID-19 pandemic causing a deep decrease in mortality [1-4].
Question 3: Discussion, please mention that the advent of COVID-19 vaccines that introduce large quantities of systemic Spike protein are likely to intensifiy the mechanisms discussed in this paper resulting in even greater progression of CKD as shown by Canney et al.
Canney reference: Canney M, Atiquzzaman M, Cunningham AM, Zheng Y, Er L, Hawken S, Zhao Y, Barbour SJ. A Population-Based Analysis of the Risk of Glomerular Disease Relapse after COVID-19 Vaccination. J Am Soc Nephrol. 2022 Nov 4:ASN.2022030258. doi: 10.1681/ASN.2022030258. Epub ahead of print. PMID: 36332971.
Answer: To clarify this issue the following text has been added:
P.5, line 214
“Furthermore, in a recent work, Canney et al. [76] showed that patients with previous glomerular disease have increased risk of relapse when exposed to a second or third dose of COVID-19 vaccine, measured by the increase in proteinuria and/or the decline of glomerular function. Since vaccines are based on the spike protein, these results indicate that spike induces a cellular response in susceptible patients’ subgroup. However, these results must be interpreted with care once the authors mentioned that there is a low absolute increased risk of relapse. Additionally, the authors did not mention if the kidney damage was transient or not. Further studies should be performed to confirm other possible adverse side effects in kidney disease patients.”
References:
1- Polack FP, Thomas SJ, Kitchin N, Absalon J, Gurtman A, Lockhart S, Perez JL, Pérez Marc G, Moreira ED, Zerbini C, Bailey R, Swanson KA, Roychoudhury S, Koury K, Li P, Kalina WV, Cooper D, Frenck RW Jr, Hammitt LL, Türeci Ö, Nell H, Schaefer A, Ünal S, Tresnan DB, Mather S, Dormitzer PR, Şahin U, Jansen KU, Gruber WC; C4591001 Clinical Trial Group. Safety and Efficacy of the BNT162b2 mRNA Covid-19 Vaccine. N Engl J Med. 2020 Dec 31;383(27):2603-2615. doi: 10.1056/NEJMoa2034577.
2- Thomas SJ, Moreira ED Jr, Kitchin N, Absalon J, Gurtman A, Lockhart S, Perez JL, Pérez Marc G, Polack FP, Zerbini C, Bailey R, Swanson KA, Xu X, Roychoudhury S, Koury K, Bouguermouh S, Kalina WV, Cooper D, Frenck RW Jr, Hammitt LL, Türeci Ö, Nell H, Schaefer A, Ünal S, Yang Q, Liberator P, Tresnan DB, Mather S, Dormitzer PR, Şahin U, Gruber WC, Jansen KU; C4591001 Clinical Trial Group. Safety and Efficacy of the BNT162b2 mRNA Covid-19 Vaccine through 6 Months. N Engl J Med. 2021 Nov 4;385(19):1761-1773. doi: 10.1056/NEJMoa2110345.
3- Watson OJ, Barnsley G, Toor J, Hogan AB, Winskill P, Ghani AC. Global impact of the first year of COVID-19 vaccination: a mathematical modelling study. Lancet Infect Dis. 2022 Sep;22(9):1293-1302. doi: 10.1016/S1473-3099(22)00320-6.
4- Voysey M, Clemens SAC, Madhi SA, Weckx LY, Folegatti PM, Aley PK, Angus B, Baillie VL, Barnabas SL, Bhorat QE, Bibi S, Briner C, Cicconi P, Collins AM, Colin-Jones R, Cutland CL, Darton TC, Dheda K, Duncan CJA, Emary KRW, Ewer KJ, Fairlie L, Faust SN, Feng S, Ferreira DM, Finn A, Goodman AL, Green CM, Green CA, Heath PT, Hill C, Hill H, Hirsch I, Hodgson SHC, Izu A, Jackson S, Jenkin D, Joe CCD, Kerridge S, Koen A, Kwatra G, Lazarus R, Lawrie AM, Lelliott A, Libri V, Lillie PJ, Mallory R, Mendes AVA, Milan EP, Minassian AM, McGregor A, Morrison H, Mujadidi YF, Nana A, O'Reilly PJ, Padayachee SD, Pittella A, Plested E, Pollock KM, Ramasamy MN, Rhead S, Schwarzbold AV, Singh N, Smith A, Song R, Snape MD, Sprinz E, Sutherland RK, Tarrant R, Thomson EC, Török ME, Toshner M, Turner DPJ, Vekemans J, Villafana TL, Watson MEE, Williams CJ, Douglas AD, Hill AVS, Lambe T, Gilbert SC, Pollard AJ; Oxford COVID Vaccine Trial Group. Safety and efficacy of the ChAdOx1 nCoV-19 vaccine (AZD1222) against SARS-CoV-2: an interim analysis of four randomised controlled trials in Brazil, South Africa, and the UK. Lancet. 2021 Jan 9;397(10269):99-111. doi: 10.1016/S0140-6736(20)32661-1.